# Modern *Acinetobacter baumannii* clinical isolates replicate inside spacious vacuoles and egress from macrophages

**Gabriela Sycz**[1⊘], **Gisela Di Venanzio**[1⊘], **Jesus S. Distel**[1], **Mariana G. Sartorio**[1], **Nguyen-Hung Le**[1], **Nichollas E. Scott**[2], **Wandy L. Beatty**[1], **Mario F. Feldman**[1]*

**1** Department of Molecular Microbiology, Washington University School of Medicine, Saint Louis, Missouri, United States of America, **2** Department of Microbiology and Immunology, The Peter Doherty Institute for Infection and Immunity, University of Melbourne, Parkville, Victoria, Australia

⊘ These authors contributed equally to this work.
* mariofeldman@wustl.edu

**Data Availability Statement:** The authors declare that data supporting the findings of this study are available within the paper and its supplemental files (Data set S1). The resulting MS data and search

## Abstract

Multidrug-resistant *Acinetobacter baumannii* infections are increasing at alarming rates. Therefore, novel antibiotic-sparing treatments to combat these *A. baumannii* infections are urgently needed. The development of these interventions would benefit from a better understanding of this bacterium's pathobiology, which remains poorly understood. *A. baumannii* is regarded as an extracellular opportunistic pathogen. However, research on *Acinetobacter* has largely focused on common lab strains, such as ATCC 19606, that have been isolated several decades ago. These strains exhibit reduced virulence when compared to recently isolated clinical strains. In this work, we demonstrate that, unlike ATCC 19606, several modern *A. baumannii* clinical isolates, including the recent clinical urinary isolate UPAB1, persist and replicate inside macrophages within spacious vacuoles. We show that intracellular replication of UPAB1 is dependent on a functional type I secretion system (T1SS) and pAB5, a large conjugative plasmid that controls the expression of several chromosomally-encoded genes. Finally, we show that UPAB1 escapes from the infected macrophages by a lytic process. To our knowledge, this is the first report of intracellular growth and replication of *A. baumannii*. We suggest that intracellular replication within macrophages may contribute to evasion of the immune response, dissemination, and antibiotic tolerance of *A. baumannii*.

## Author summary

*Acinetobacter baumannii* is a nosocomial pathogen that causes multiple types of infection. This bacterium has an alarming predisposition to acquire multi-drug resistance, and infections associated with these strains are linked to greater morbidity and mortality. Therefore, novel antibiotic-sparing treatments to combat these *A. baumannii* infections are urgently needed. The development of these interventions would benefit from a better understanding of the mechanisms employed by *A. baumannii* to cause infection. *A. baumannii* is regarded as an extracellular opportunistic pathogen. However, research on

results have been deposited into the PRIDE ProteomeXchange Consortium repository[47,48] and can be accessed with the identifier PXD024736 using the Username: reviewer_pxd024736@ebi.ac. uk and Password: YS5qjHcG.

**Funding:** MFF was supported by grants from the National Institute of Allergy and Infectious Diseases (grant R01AI144120). The funders had no role in study design, data collection and analysis, decision to publish, or preparation of the manuscript.

*Acinetobacter* has largely focused on common lab strains that have been isolated several decades ago. These strains exhibit reduced virulence when compared to recently isolated clinical strains. In this work, we demonstrate that a subset of modern *A. baumannii* clinical isolates persist and replicate inside macrophages within spacious vacuoles, and identify bacterial factors involved in the process. We propose that replication inside macrophages may contribute to the evasion of the immune response and bacterial dissemination. Antibiotics cannot penetrate the macrophages, and therefore our findings contribute to explain the extremely high tolerance of *A. baumannii* to antibiotics. Our work opens new avenues for the development of novel therapeutic interventions against this worrisome human pathogen.

## Introduction

Infections caused by the bacterial pathogen *Acinetobacter baumannii* are a global public health risk since multidrug resistance (MDR) increasingly limits effective therapeutic interventions [1,2]. The emerging success of this pathogen has led the World Health Organization (WHO) and the CDC to categorize carbapenem-resistant *A. baumannii* as a top priority for the research and development of new antibiotics[3,4]. However, compared to other opportunistic pathogens, our knowledge about the pathobiology of *Acinetobacter* is rather limited.

Research on *Acinetobacter* has largely focused on common lab strains that have been "lab-domesticated" over ~50 years. These non-MDR strains exhibit reduced virulence when compared to recently isolated clinical strains, indicating that classical lab strains lack virulence factors expressed by current-day strains[5,6]. Consequently, little is known about the virulence factors and molecular pathways governing modern *A. baumannii* infections. Although *A. baumannii* is best known as a respiratory pathogen, infection of various anatomical sites is also common. For example, urinary tract infections (UTI) account for about ~20% of *A. baumannii* infections worldwide[5]. We recently developed a murine model of *A. baumannii* catheter-associated urinary tract infection (CAUTI)[5]. Using this model, we compared the behavior of two urinary *A. baumannii* isolates, ATCC19606 (19606) and UPAB1. 19606 is a urinary isolate obtained in 1967, and it has been routinely used to study *A. baumannii* virulence. UPAB1 is a modern MDR UTI isolate, collected from a female patient in 2016. We demonstrated that, unlike 19606, which was rapidly cleared from infected mice, UPAB1 was able to establish early implant and bladder colonization[5]. These findings suggested that "old" and "new" strains have a different repertoire of virulence or fitness factors, which can influence the outcome of the infection. Interestingly, UPAB1 carries a large conjugative plasmid (LCP), pAB5, which controls the expression of multiple virulence factors. Remarkably, pAB5 impacts the pathogenicity of UPAB1 in the CAUTI and acute pneumonia murine models[5].

A few reports have suggested that classical *A. baumannii* strains are able to persist within epithelial cells. It has been suggested that the intracellular bacterial cells are contained by membrane-bound vacuoles, likely of autophagic origin, and subsequently eliminated by the intracellular vesicular pathway[7–10]. In this work, we demonstrate that unlike 19606, several modern *A. baumannii* clinical isolates, including UPAB1, persist and, remarkably, replicate inside macrophages within spacious vacuoles containing up to more than 30 bacteria. We show that intracellular replication of UPAB1 is dependent on pAB5 and a functional Type I Secretion System (T1SS). Finally, we also show that UPAB1 escapes from the infected macrophages by a lytic process. To the best of our knowledge, this is the first report of intracellular growth and replication of *A. baumannii*.

## Results

### UPAB1 but not 19606 replicates in spacious intracellular vacuoles within macrophages

Macrophages play an important role in early host defense against *A. baumannii*. Indeed, depletion of macrophages enhances sepsis and severity of *A. baumannii* infection [11]. Macrophages can efficiently control the replication of *A. baumannii* and kill the most common lab strains, such as ATCC 17978 and 19606, which have been isolated more than half a century ago. However, whether modern clinical isolates can be efficiently phagocytosed and eliminated by macrophages has not been investigated. To this end, we tested the ability of the modern *A. baumannii* isolate UPAB1 to resist macrophage-killing. J774A.1 macrophages were infected at an MOI of 10 and visualized by TEM. We found that UPAB1-infected J774A.1 macrophages possessed spacious vacuoles containing up to 15 bacteria per vacuole (Fig 1A). Hereafter, we refer to these structures as *A*.*baumannii*-Containing Vacuoles (ACVs). A double membrane was visible in some of the ACVs, which is compatible with an autophagic origin (Fig 1B, right panel). This result is in agreement with previous reports showing that *A. baumannii* lab strains are contained within autophagic ACVs in epithelial cells [12]. CFU enumeration of intracellular bacteria (Fig 2A) showed that UPAB1 can replicate intracellularly between 2 to 6 h p.i. In contrast, the lab strain 19606 was efficiently eliminated and no intracellular replication was detected (Fig 2A). Visualization of intracellular bacteria by confocal microscopy demonstrated the presence of small ACVs at 2.5 h p.i and some large ACVs, containing more than 20 bacteria per vacuole, at 5 and 7 h p.i. (Figs 2B and S1). Indeed, the number of bacteria per ACV

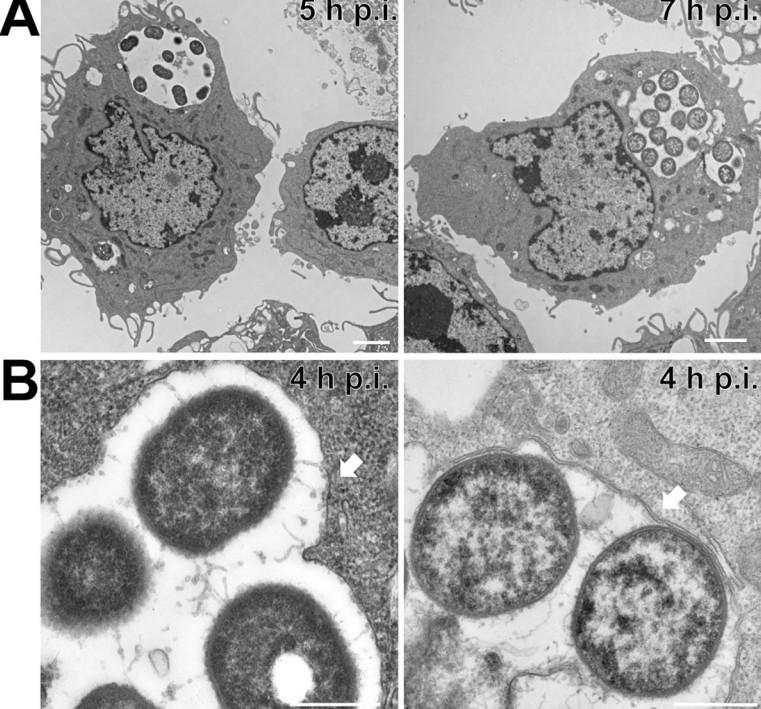

**Fig 1. UPAB1 resides in spacious vacuoles within macrophages.** Transmission electron microscopic images of J774A.1 macrophages infected with UPAB1 **(A)** at 5 (left panel) and 7 h p.i. (right panel) Bars: 2 μm. **(B)** The UPAB1 containing vacuole can have a single (left panel, white arrow) or double membrane (right panel, white arrow). Bars: 500 nm.

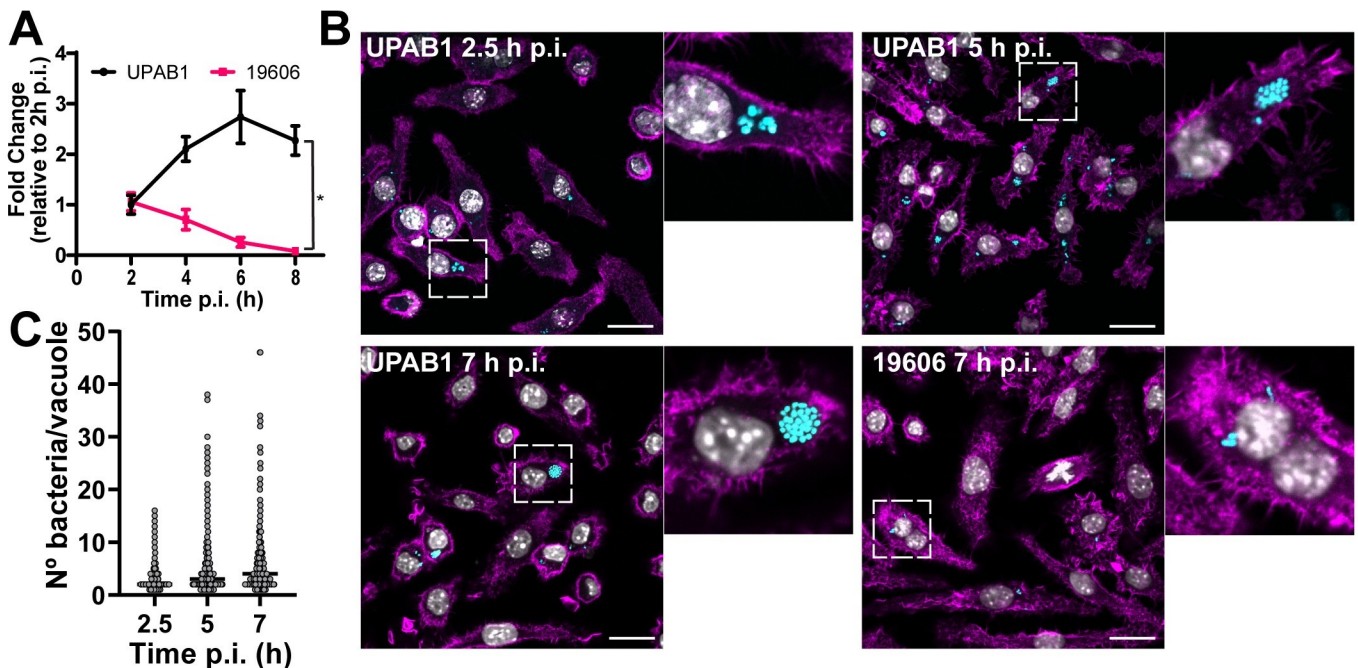

**Fig 2. UPAB1 but not 19606 replicates in macrophages. (A)** J774A.1 macrophages were infected with the indicated strains, and after phagocytosis, extracellular bacteria were eliminated with antibiotics. After different time points, total numbers of intracellular CFU were determined. *$p < 0.05$. **(B)** Representative images of infected cells at the indicated time points. Samples were stained for cell nuclei (grey), actin (magenta) and *Acinetobacter* GFP (cyan). Insets show a higher magnification of the area indicated by the white box. Individual channels are presented in S3 Fig. Bars: 20 μm. **(C)** Quantification of bacteria per vacuole at different times p.i. Each symbol represents an individual vacuole, horizontal line represents the median. At least 200 infected cells were analyzed for each time point for three independent experiments.

increased with time (Figs 2B, 2C and S1 and S2). We were able to detect vacuoles with up to 30 to 40 intracellular UPAB1 between 5 to 7 h p.i. Almost no intracellular bacteria were observed in macrophages infected with 19606 at similar time points (Figs 2B, right lower panel, and S2). Together, these experiments demonstrate that UPAB1 is able to infect, persist and replicate intracellularly in J774A.1 macrophages.

## UPAB1 ACVs are LC3-negative

Autophagy is a well-established intracellular mechanism by which the eukaryotic cells eliminate unwanted components, including invading pathogens[13]. However, some intracellular pathogens are able to grow in vacuoles with autophagic features, like *Serratia marcescens* and *Coxiella burnetii* [14,15]. Several studies employing non-replicative *A. baumannii* lab strains have proposed that upon infection, the autophagic pathway is activated in host cells, leading to an accumulation of LC3-positive vacuoles[8]. Although many alternative molecular pathways have been described in the autophagic process, one common characteristic of the autophagic vacuoles is a double membrane and the presence of the protein LC3 in both membranes[16]. As some of the UPAB1-containing vacuoles exhibited a double membrane, we investigated if these vacuoles contained LC3 (S3 Fig). Uninfected macrophages showed a smooth cytoplasmic distribution of LC3, characteristic of non-autophagic processes (S3A Fig). *Serratia marcescens* RM66262 vacuoles were LC3-positive at 2 h p.i. (S3B Fig), as previously described[14]. However, no LC3-positive vacuoles (S3C Fig) were visualized in cells infected with UPAB1 at all the observed time points (1, 2.5 and 5 h p.i.), suggesting that replicative ACVs follow a non-canonical intracellular pathway.

### UPAB1 also replicates in THP-1 cells and bone-marrow-derived macrophages (BMDM)

Large ACVs were also observed in human THP-1 cells (Figs 3, left panel and S4) and, remarkably, in murine BMDM (Figs 3, right panel and S2 and S4), indicating that intracellular replication of UPAB1 is a phenomenon that occurs in multiple host cells and does not depend on a permissive host cell. For the remainder of the study, we will employ J774A.1 macrophages as host cells.

### A subset of MDR clinical isolates replicate inside macrophages

Lab-domesticated strains generally exhibit reduced virulence in animal models when compared to recently isolated clinical strains, a possible indication that classical lab strains lack virulence factors expressed by current-day strains[17]. We next tested if other recent MDR clinical isolates were able to replicate in macrophages (Fig 4). Although at different rates, many of these clinical isolates also replicated in J774A.1 macrophages (Fig 4A). These strains were also contained in large ACVs (Figs 4B and S5), demonstrating that intracellular replication is not a unique feature of UPAB1. We found that AbCAN2[18], a recent bone isolate was not able to replicate and was quickly eliminated by J774A.1 macrophages. These experiments demonstrate that a subset of modern clinical isolates replicate within macrophages.

### UPAB1 escapes the macrophages by a lytic process

Our results show that intracellular CFUs of UPAB1 in J774A.1 peak at 4–6 h p.i., and then decrease at later time points (Fig 2A). This decline may be the result of bacterial death inside the vacuoles. Alternatively, the reduction in the CFUs of intracellular bacteria could reflect the egress of bacteria from the infected macrophage to the extracellular media. To investigate these hypotheses, we carried out a modified antibiotic protection assay. In this assay, antibiotics were added 1 h p.i. to eliminate extracellular non-phagocytosed bacteria. At different time points, the antibiotic-containing medium was replaced by fresh cell culture medium without antibiotics. After a 20 min incubation, the culture supernatants from each well were recovered and the number of released bacteria were quantified. The ratio of released bacteria in relation to the number of intracellular bacteria corresponding to each time interval is shown in Fig 5A. Our data suggests that intracellular UPAB1 cells egress from the macrophages. To confirm this result, we inspected the infected macrophages with UPAB1-GFP through live-video

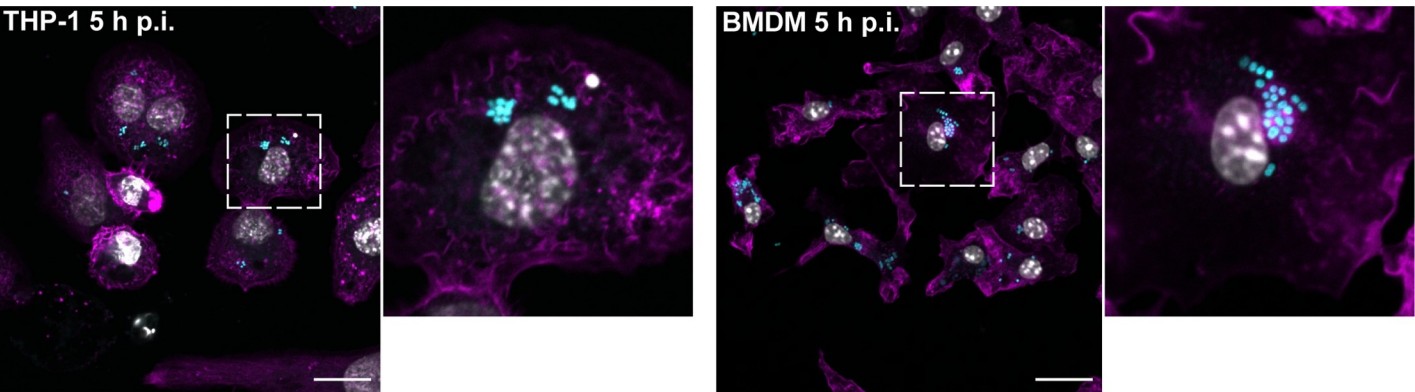

**Fig 3. UPAB1 replicates in multiple host cell lines.** Representative images of infected THP-1 (left panels) or BMDM (right panels) cells at 5 h p.i. Samples were stained for cell nuclei (grey), actin (magenta) and UPAB1 GFP (cyan). Insets show a higher magnification of the area indicated by the white box. Individual channels are presented in S4 Fig. Bars: 20 μm.

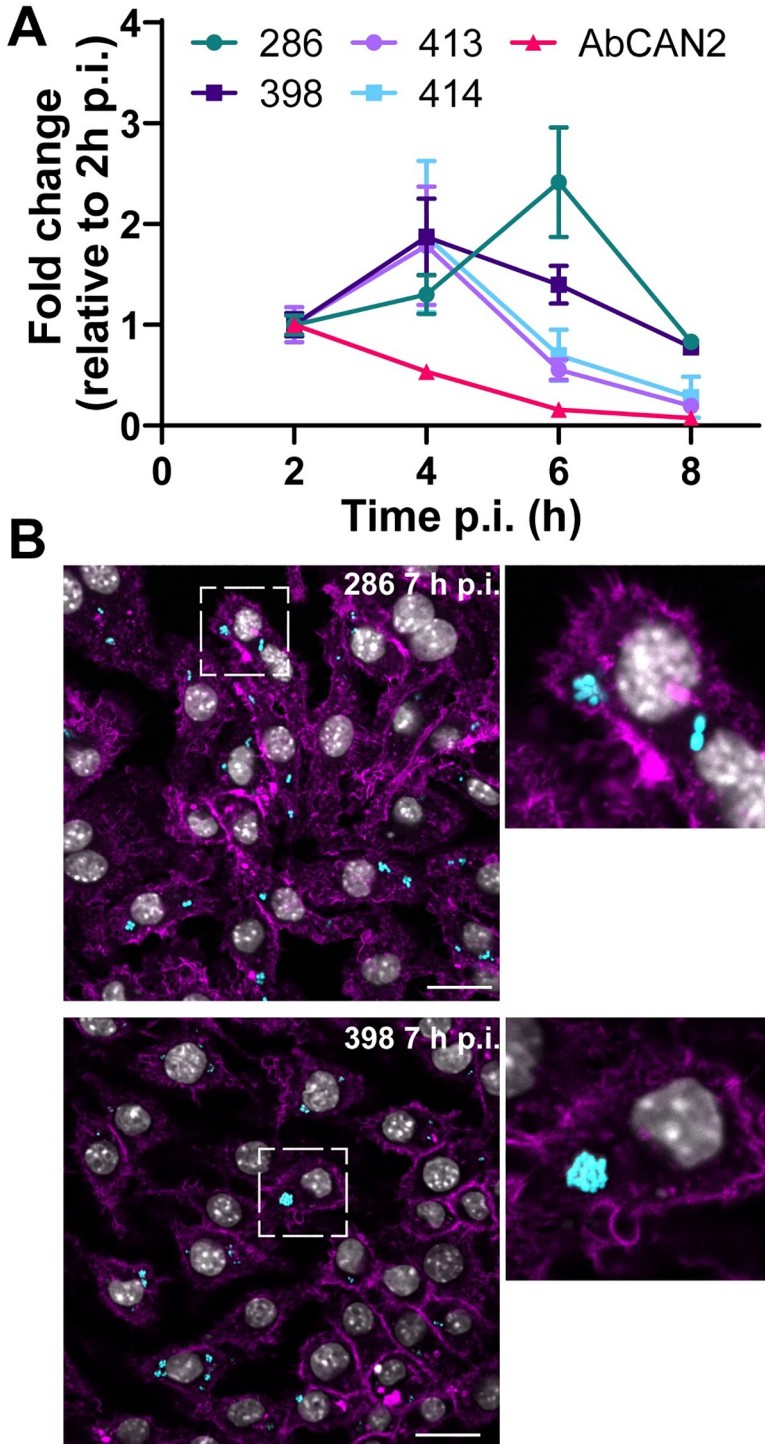

**Fig 4. Recent clinical isolates replicate in macrophages. (A)** J774A.1 macrophages were infected with the indicated strains and, after phagocytosis, extracellular bacteria were eliminated with antibiotics. After different time points, total numbers of intracellular CFU were determined. Data represent mean and standard deviation values for 3 independent experiments. **(B)** Representative images of infected cells with the indicated strains. Samples were stained for cell nuclei (grey), actin (magenta) and *Acinetobacter* GFP (cyan). Insets show a higher magnification of the area indicated by the white box. Individual channels are presented in S5 Fig. Bars: 20 μm.

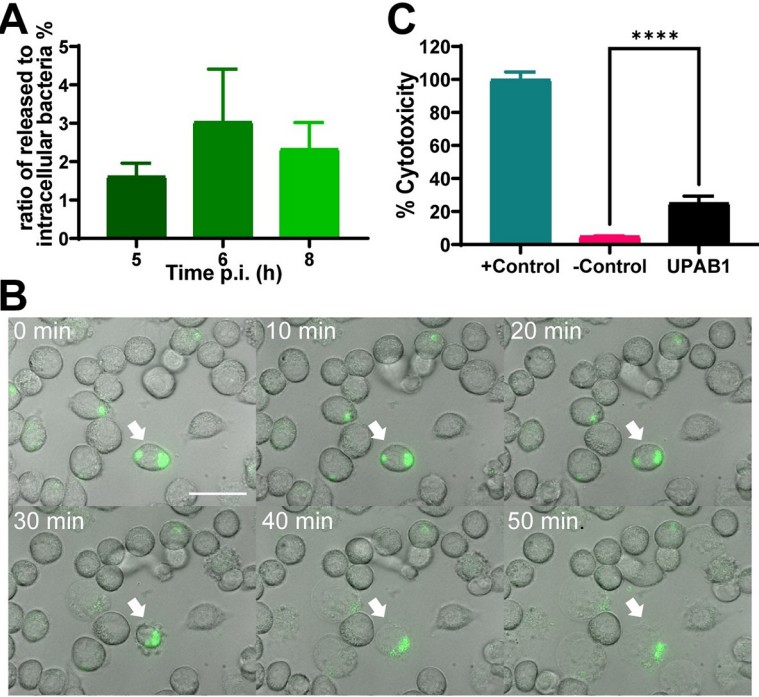

**Fig 5. UPAB1 escapes the macrophages by a lytic process. (A)** J774A.1 macrophages were infected with UPAB1. **(A)** At different time points, antibiotic containing medium was replaced by antibiotic-free medium. CFU in supernatants were determined and the ratio of released to intracellular bacteria was calculated. Data represent mean and standard deviation values for 3 independent experiments. **(B)** Images from time-lapse microscopy showing UPAB1 egress from the invaded macrophages. J774A.1 cells were infected with UPAB1 GFP (green fluorescence). At 7 h p.i. images were taken every 10 min and analyzed with ImageJ software. The white arrow indicates the infected cell. Bars: 20 μm. **(C)** LDH activity in the supernatant of infected macrophages was measured at 24 h p.i. Percentage of cytotoxicity was calculated as the activity of released LDH relative to total LDH activity. The mean ± S.D. for two independent experiments is shown. **** $p < 0.0001$.

microscopy. We observed bacteria egressing the macrophages through a process that lysed the host cells (Fig 5B). Lactate dehydrogenase (LDH), a soluble cytosolic enzyme present in eukaryotic cells, is released into the culture medium upon damage to the plasma membrane [19]. The increase of LDH activity in culture supernatant is proportional to the number of damaged cells. No LDH activity was detected at 8 or 12 h p.i. However, we detected a significant increase in LDH activity in macrophages infected with UPAB1 at 24 h p.i. compared to non-infected macrophages (Fig 5C). Together, these results indicate that UPAB1 can survive and replicate inside macrophages and then escape from these cells through a lytic process. Although the mechanism for these processes will require further studies, it is tempting to speculate that UPAB1, like other bacteria that replicate and escape from the vacuole, manipulates the host cell through the activity of secreted proteins.

## UPAB1 replication in macrophages relies on pAB5

LCPs are a family of large conjugative plasmids present in many *A. baumannii* strains[20]. In previous work, we showed that pAB5, the LCP carried by UPAB1, greatly impacts virulence as its presence is required for the establishment of infection in a CAUTI model but detrimental for virulence in an acute respiratory murine model[5]. Thus, we investigated if the presence of pAB5 also impacts the ability of UPAB1 to replicate in macrophages. The strain cured of its plasmid (UPAB1p-) was unable to replicate in J774A.1 macrophages (Fig 6A), and intracellular

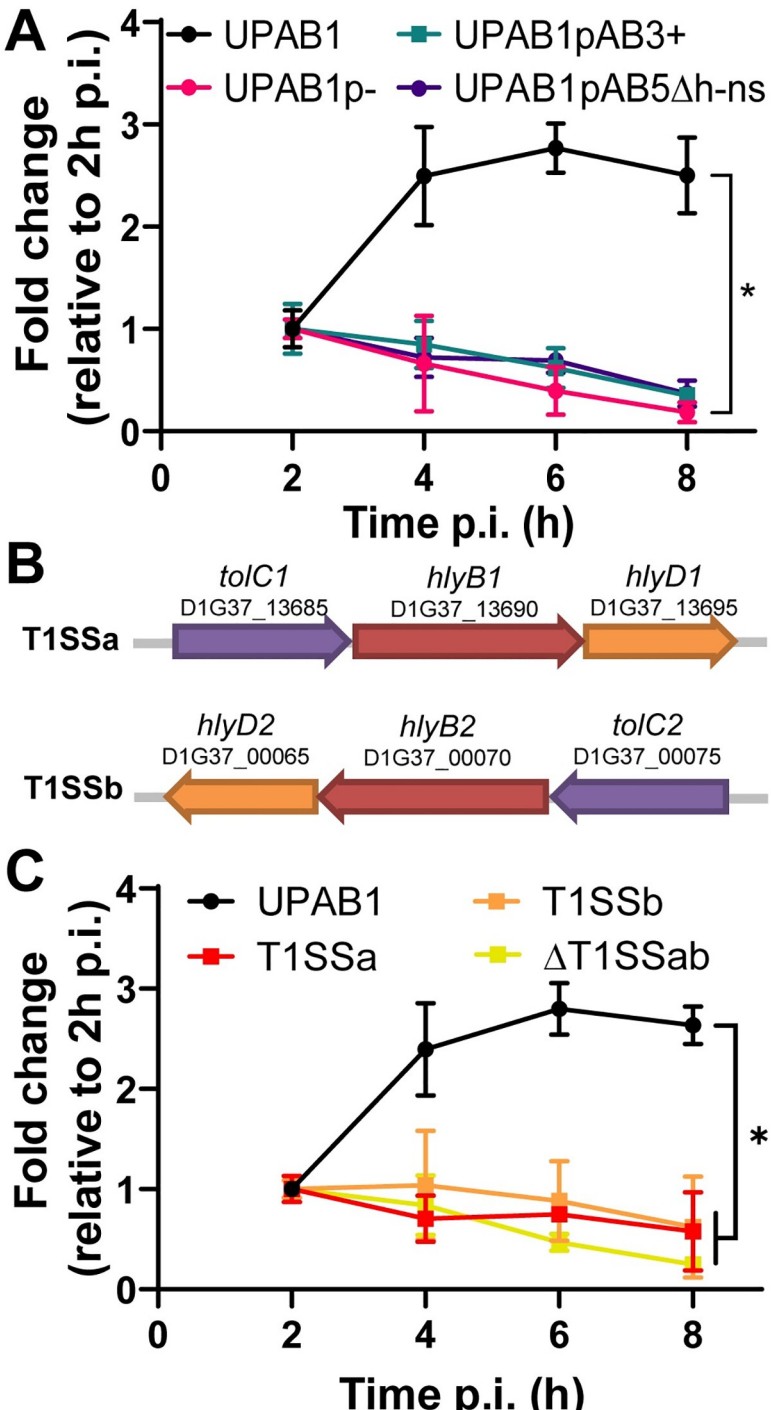

**Fig 6. pAB5 and T1SS are required for UPAB1 intracellular replication. (A)** J774A.1 macrophages were infected with the indicated strains. UPAB1p-, UPAB1pAB3+ and UPAB1pAB5Δh-ns strains were not able to replicate. Data represents mean and standard deviation values for 3 independent experiments. $^*p < 0.05$. **(B)** Genetic organization of the T1SSa and T1SSb loci. **(C)** Individual and double T1SS mutants have a defect in intracellular replication in J774A.1 macrophages. Data represents mean and standard deviation values, for 3 independent experiments. $^*p < 0.05$.

CFUs quickly decreased after phagocytosis, indicating that UPAB1p- is efficiently eliminated by macrophages. We have previously shown that LCP controls the expression of the type VI secretion system (T6SS) [21]. We also found that pAB5 modulates the expression of other chromosomally-encoded factors. For example, pAB5 represses the synthesis of poly-N-acetyl-β-(1–6)-glucosamine (PNAG), a surface polysaccharide that plays an important role in biofilm formation *in vitro*, and reduces the expression of two chaperone-usher pili, CUP1 and CUP2, which are required for adhesion to fibrinogen and possibly other host receptors during infection[5]. We have recently showed that a pAB5-encoded H-NS transcriptional regulator represses the synthesis of the exopolysaccharide PNAG and other cell surface components [22]. Unlike pAB5, the closely related plasmid pAB3, does not repress PNAG synthesis [5]. Thus, to gain further insight about the roles of LCPs in UPAB1 replication, we tested the strains UPAB1 carrying pAB3 (UPAB1pAB3+) and UPAB1pAB5Δh-ns. Both strains were incapable of replicating inside J774A.1 macrophages and were quickly eliminated by the host. These results demonstrate that pAB5 enables intracellular replication of UPAB1 in a H-NS-dependent manner.

Most bacteria that are able to survive and replicate inside macrophages secrete proteins via secretion systems, commonly T3SS, T4SS, T6SS, or T7SS[23]. These effectors allow the bacteria to manipulate host signaling and trafficking pathways to create a replicative niche within the macrophage. *Acinetobacter* spp. do not encode T3SS or T4SS, but carry T1SS, T2SS and T6SSs [24]. However, the T6SS in *Acinetobacter* appears to be only involved in interbacterial competition, as many recent clinical isolates do not carry a functional T6SS[18]. Our previous proteomic and RNAseq experiments showed that pAB5 increases the expression of a T1SS-secreted agglutinin RTX toxin (D1G37_00080), and an adjacently-encoded protein, OmpA[5]. This observation prompted us to investigate a possible role for the T1SS in the intracellular replication of UPAB1.

## T1SS is required for UPAB1 intracellular replication

The T1SS is a highly conserved secretion system employed by pathogenic Gram-negative bacteria. This secretion machinery is composed of an ATP-binding cassette transporter anchored in the inner membrane (HlyB), a periplasmic adaptor protein (HlyD) and a third component localized in the outer membrane (TolC). Together, these elements act to facilitate the secretion of unfolded effectors from the cytoplasm to the extracellular milieu[25]. UPAB1 encodes two T1SS clusters, which we named T1SSa and T1SSb (Fig 6B). We generated ΔT1SSa, ΔT1SSb and ΔT1SSab mutant strains and analyzed their replication within macrophages. We found that all three T1SS deficient strains were unable to replicate inside J774A.1 macrophages (Fig 6C).

To identify putative T1SS-dependent effectors involved in survival and replication within macrophages, we employed a differential proteomics approach. A comparative secretome analysis between WT UPAB1 and the T1SSa and T1SSb mutant strains was performed. Table 1 lists the secreted proteins diminished in ΔT1SSa and ΔT1SSb compared to WT. We were able to identify multiple T1SS effector candidates. T1SS effectors BapA and RTX2, two common T1SS effectors, were repressed in ΔT1SSa. Neither of these proteins were repressed in ΔT1SSb, suggesting that T1SSa is the preferred secretion system employed for the export of these two proteins. Our analysis also identified additional potential T1SS effectors, including proteases, phosphatases, glycosidases, and a putative invasin (Table 1). In other bacteria, orthologs of this invasin are required to induce bacterial entry and simultaneously to block the generation of reactive oxygen species (ROS) by host monocytes and macrophages[26]. Although the identification of the T1SS effectors directly involved in intracellular replication is

**Table 1. Putative T1SS-dependent effectors.**

| T-test Difference mut_WT | Locus tag | Predicted function |
|---|---|---|
| | **ΔT1SSa** | |
| -10.5 | D1G37_01945 | BapA |
| -9.1 | D1G37_04720 | OmpA |
| -5.7 | D1G37_05375 | PilY1 |
| -5.4 | D1G37_18005 | Hypothetical protein |
| -5.2 | D1G37_15450 | Phosphodiester glycosidase family protein |
| -5.0 | D1G37_13335 | Gly zipper domain/ toxic amyloid oligomer |
| -4.7 | D1G37_05180 | Secreted serine peptidase |
| -4.2 | D1G37_05495 | type IV pilus secretin PilQ |
| -4.0 | D1G37_12590 | Curli production component |
| -3.9 | D1G37_01055 | Colicin synthesis protein, bactericidal |
| -3.8 | D1G37_10435 | Bacteriocin resistance protein, peptidase |
| -3.8 | D1G37_03120 | RTX2 |
| -3.8 | D1G37_16210 | Phosphatase |
| -3.4 | D1G37_05480 | Pili operon hypothetical protein |
| -3.3 | D1G37_12980 | Hypothetical protein, exported |
| | **ΔT1SSb** | |
| -9.9 | D1G37_05375 | PilY1 |
| -8.0 | D1G37_05445 | PilA |
| -3.9 | D1G37_04400 | invasin/ igG like |
| -3.7 | D1G37_00110 | Cell surface protein Ata |
| -3.6 | D1G37_10430 | FilA |
| -3.4 | D1G37_11695 | Transferrin binding protein |
| -3.4 | D1G37_00700 | Hypothetical protein |

Cut-off was defined at -3 by a student T-test difference between mutant and WT. Protein localization prediction was performed employing PSORT and presence of a signal peptide was analyzed by SignalP.

beyond the scope of this work, our experiments demonstrate that T1SSs play a role in the process, and our MS analysis identified several interesting candidates that will be the subject of future studies.

## Discussion

Although intracellular survival of *A. baumannii* inside vacuoles has been reported, the capacity of this bacterium to replicate intracellularly has not been documented. Here, we show that modern clinical isolates, such as UPAB1, can replicate inside spacious vacuoles within macrophages. Previous reports have shown that non-replicating *A. baumannii* can induce autophagy in epithelial cells and macrophages and indicated that autophagy is required for clearance of intracellular bacteria[7–10]. TEM predominantly exhibited double-membrane enclosed vacuoles, which is indicative of an autophagic origin for the ACV. However, we found that ACVs formed by UPAB1 are LC3-negative, suggesting that these replicative ACVs do not follow a canonical autophagy pathway. It remains to be determined whether these observations are the result of LC3 degradation induced by UPAB1 or an indication that UPAB1 evades the canonical autophagy pathway before LC3 is recruited to the autophagosome. For example, in fibroblasts double membrane vacuoles can be formed without recruitment of the ATG conjugation system, leading to a complete absence of LC3 recruitment to the vacuole[27]. Moreover,

intracellular pathogens manipulate the canonical host vesicular pathways, including autophagy, to create unique replicative niches[28]. The complete characterization of the replicative ACV will be the subject of future studies.

During the revision of this work, the Salcedo group reported that a subset of modern clinical isolates replicates within non-autophagic ACVs in epithelial cells[29]. In agreement with our results, large ACVs containing up to 50 bacteria were visualized in multiple non-phagocytic cells. However, some apparent differences between the ACVs formed in epithelial cells and macrophages were detected. Mainly, the replicating *Acinetobacter* strains were contained in single membrane vacuoles and were observed after 24 h p.i. It remains to be elucidated if these discrepancies are due to the use of different host cells and/or the particular bacterial strains tested.

Several MDR *A. baumannii* strains carry LCPs. These plasmids control the expression of diverse chromosomally-encoded genes. Specifically, pAB5, the LCP of UPAB1, differentially regulates the expression of various features, such as PNAG synthesis and CUP pili, directly impacting host virulence. We found that UPAB1p-, UPAB1pAB5Δh-ns, and UPAB1pAB3 + are unable to replicate within macrophages, suggesting that genetic elements regulated by pAB5 are likely involved in facilitating intracellular replication. We previously reported that pAB5 upregulates putative T1SS effectors[5]. UPAB1 carries two T1SS loci, and our data revealed that mutants in both systems are defective in intracellular replication. Although the T3SS, T4SS, and T6SS are most commonly involved in manipulation of intracellular pathways, the T1SS is involved in invasion and intracellular replication in other bacterial pathogens, such as *Legionella pneumophila*, *Ehrlichia chaffeensis*, *Francisella novicida* and *Orientia tsutsugamushi*[30–33]. Our proteomic analysis identified several proteins whose secretion is significantly reduced in the individual T1SS mutants. However, since deletion of the T1SS in *A. nosocomialis* resulted in the downregulation of other secretion systems[34], the assignment of these proteins as T1SS effectors awaits further validation. Nevertheless, these experiments uncover several interesting, secreted proteins that may play a role in intracellular replication of UPAB1. Future work will focus on investigating these putative effectors and their possible role in the intracellular lifestyle of *A. baumannii*.

Depletion of macrophages enhances sepsis and severity of disease, demonstrating that macrophages play an important role in eradicating *A. baumannii* infection[11]. Therefore, survival and replication of *A. baumannii* in this type of cells may be a previously underappreciated and important feature of this bacterium. We have also shown that UPAB1 can escape the macrophages by a lytic mechanism. *A. baumannii* could subsequently be phagocytosed by other macrophages and re-start the infection process. Thus, we propose that by replicating inside vacuoles in macrophages, *A. baumannii* evades the immune response and disseminates, resulting in hypervirulent phenotypes. Furthermore, it is tempting to speculate that antibiotics cannot reach the lumen of the ACV, which may contribute to tolerance of *A. baumannii* to antibiotics. Identifying the bacterial factors involved in the intracellular cycle of MDR *A. baumannii* may enable the design of new therapies to combat this pathogen.

## Materials and methods

### Bacterial strains and culture conditions

The bacterial strains used in this study are listed in S1 Table. Unless otherwise noted, strains were grown in lysogeny broth (LB) liquid medium under shaking conditions (200 rpm), or in LB-agar at 37˚C. When required, antibiotics were used at the following concentrations: zeocin 50 μg/ml, kanamycin 30 μg/ml, gentamicin 15 μg/ml, and chloramphenicol 15 μg/ml.

## DNA manipulation

DNA manipulations were performed according to standard techniques. PCR reactions were performed using Phusion High Fidelity DNA polymerase (New England Biolabs), according to the manufacturer's instructions. Genetic manipulations were carried out in *Escherichia coli* Stellar competent cells (Clontech). Primers and plasmids used in this study are listed in S2 and S3 Tables, respectively.

The *gfp* gene was PCR amplified from pBAV1K-t5-gfp[35] and cloned into pUC18T-mini-Tn7T-Zeo by In-Fusion Cloning (Takara). The pUC18T-mini-Tn7T-Zeo-GFP vector was conjugated to *Acinetobacter* as previously described[36]. All transconjugant strains were confirmed by PCR analysis and fluorescence microscopy.

## Construction of *A. baumannii* mutant strains

Mutants were constructed as described previously[37]. Briefly, a FRT site-flanked zeocin resistance cassette was amplified from a variant of pKD4[38] with primers harboring 18–25 nucleotides of homology to the flanking regions of the targeted gene to delete. Upstream and downstream flanking regions were also amplified, and fragments were assembled by overlap extension PCR. This PCR product was electroporated into competent UPAB1 carrying pAT04, which expresses the RecAB recombinase[37]. Mutants were selected with zeocin, and integration of the resistance marker was confirmed by PCR. To remove the zeocin resistance cassette, electrocompetent mutants were transformed with pAT03 plasmid, which expresses the FLP recombinase. Zeocin-sensitive clones of clean deletion mutants were obtained. All mutant strains were confirmed by antibiotic resistance profile, PCR, and sequencing.

## Cell culture conditions

J774A.1 (ATCC TIB-67) mouse macrophage cell line was cultured in Dulbecco's Modified Eagle Medium (DMEM) High Glucose (Hyclone, SH30022.01) supplemented with 10% inactivated Fetal Bovine Serum (FBS, Corning) at 37°C and 5% $CO_2$. THP-1 (ATCC TIB-202) human monocyte cell line was cultured in RPMI-1640 medium (Sigma, R8758) supplemented with 10% FBS, 0.01% MEM Non-Essential Amino Acids (Corning, 25-025-Cl), and 50 μM 2-mercaptoethanol at 37°C and 5% $CO_2$. For the infection experiments, the THP-1 monocytes were differentiated to macrophages by supplementing the medium with 40 ng/ml of phorbol 12-myristate 13-acetate (PMA, AdipoGen Life Sciences) 48 h prior to the infection.

The bone marrow derived macrophages (BMDM) were obtained from wild-type BALB/c mice as described previously[39] and cultured in DMEM supplemented with 10% FBS at 37°C and 5% $CO_2$. For the infection experiments, the BMDM were incubated with 10 ng/ml γ-INF overnight prior to the infection.

## Antibiotic protection assay

For the intracellular replication assays, J774A.1 cells were cultured in 48 well plates 12–16 h before the experiment ($3x10^5$ cells/well). *A. baumannii* strains were grown overnight in LB at 37°C under shaking conditions. Bacterial cultures were normalized to $OD_{600} = 1$, washed once with PBS, and an appropriate volume was added to each well to reach an MOI of 10. Plates were centrifuged 10 min at 200 x g and incubated for 1 h at 37°C and 5% $CO_2$. Cells were washed three times with PBS and subsequently incubated with medium supplemented with 100 μg/ml gentamicin and 400 μg/ml kanamycin to kill extracellular bacteria. At 2 h post infection (p.i.) the concentration of antibiotics was decreased to 30 μg/ml gentamicin and 200 μg/ml kanamycin.

At 2, 4, 6, and 8 h p.i., cells were washed and lysed with 0.1% Triton X-100. Colony forming units (CFUs) were determined by serial dilution.

To determine the number of bacteria that egress from the infected cells, the antibiotic containing medium was replaced with antibiotic-free medium. At the indicated time points and after 20 min incubation, the supernatant was recovered and serially diluted to determine CFU. The ratio of released bacteria in relation to the number of intracellular bacteria corresponding to each time point was then calculated.

## Immunofluorescence staining

For LC3B staining, samples were fixed with ice cold 100% methanol for 5 min at room temperature, permeabilized with 1% saponin for 20 min, and blocked with 1% PBS-BSA for 1 h. The glass cover slips were incubated with the primary antibody rabbit anti-LC3 (Sigma) at a 1:100 dilution for 3 h at room temperature, then washed 3 times with 1X PBS and incubated with the secondary antibody goat anti-rabbit Alexa 594 (Invitrogen) at a 1:100 dilution and DAPI for 1 h at room temperature. Afterwards, samples were washed with PBS and mounted with Pro-Long Glass Antifade Mountant solution (Invitrogen).

## Confocal microscopy

Infected cells were analyzed with a Zeiss LSM880 laser scanning confocal microscope (Carl Zeiss Inc.) equipped with 405nm diode, 488nm Argon, 543nm HeNe, and 633nm HeNe lasers. A Plan-Apochromat 63X (NA 1.4) DIC objective and ZEN black 2.1 SP3 software were used for image acquisition.

## Live imaging

$8x10^5$ J774A.1 cells were plated in a 10 mm Glass Bottom Culture 35 mm petridish (MATEK corporation, P35G-0-14-C) 12–16 h prior to the infection. On the following day, a gentamicin-kanamycin protection assay was performed using UPAB1 GFP at an MOI = 10 as described above. At 2 h p.i., cells were incubated in DMEM without phenol red (Hyclone) supplemented with 10% FBS, 30 μg/ml gentamicin, and 200 μg/ml kanamycin. Imaging started at 3.5 h p.i. and pictures were taken every 10 or 20 min.

## Time lapse microscopy

Live images were acquired on a Zeiss Observer Z1 inverted microscope equipped with a temperature-controlled $CO_2$ incubation chamber. Fluorescence images were acquired with illumination from a Colibri 7 LED light source (Zeiss) and ORCA-ER digital camera (Hammamatsu Photonics, Japan). A Plan-Apochromat 63X (NA 1.4) Phase 3 objective and ZEN blue 2.5 software were used for image acquisition.

## Transmission electron microscopy

For the Transmission Electron Microscopy (TEM) assay, J774A.1 cells were cultured in 24 well plates 12–16 h before the experiment ($2.5x10^5$ cells/well). The next day, an antibiotic protection assay was performed using UPAB1 GFP at an MOI = 10. At 4, 5, and 7 h p.i., the cells were washed, detached with a scraper, and pooled together. Cells were fixed in 2% paraformaldehyde/2.5% glutaraldehyde (Polysciences Inc., Warrington, PA) in 100 mM sodium cacodylate buffer pH 7.2 for 1 h at room temperature and subsequently transferred to 4˚C overnight. Samples were washed in sodium cacodylate buffer at room temperature and postfixed in 1% osmium tetroxide (Polysciences Inc.) for 1 h. Samples were then rinsed extensively in distilled

water prior to bloc staining with 1% aqueous uranyl acetate (Ted Pella Inc., Redding, CA) for 1 h. Following several rinses in distilled water, samples were dehydrated in a graded series of ethanol and embedded in Eponate 12 resin (Ted Pella Inc.). Sections of 95 nm were cut with a Leica Ultracut UCT ultramicrotome (Leica Microsystems Inc., Bannockburn, IL), stained with uranyl acetate and lead citrate, and viewed on a JEOL 1200 EX transmission electron microscope (JEOL USA Inc., Peabody, MA) equipped with an AMT 8 megapixel digital camera and AMT Image Capture Engine V602 software (Advanced Microscopy Techniques, Woburn, MA).

## Cytotoxicity assay

Lactate dehydrogenase (LDH) activity was measure with the CytoTox-One Homogeneous Membrane Integrity Assay kit (Promega), according to the manufacturer directions. Enzymatic activity, as a signal of damage of plasma membrane, was measured from the supernatants of infected cells at 8, 12 and 24 h p.i. Non-infected cells were included as a negative control, and cells treated with 0.1% Triton X-100 were the 100% lysis control.

## Secreted protein enrichment for quantitative proteomics analysis

UPAB1 and derivative strains were grown overnight in 5 ml of M9 minimal medium supplemented with 0.2% casamino acids (M9CAA) and subsequently diluted to $OD_{600} = 0.05$ in 100 ml of fresh M9CAA. Cultures were incubated to mid-log phase and centrifuged at 15,000 x g for 2 min. Supernatants were filter-sterilized with Steriflip vacuum-driven filtration devices (Millipore), concentrated with an Amicon Ultra (Millipore) concentrator with a 10-kDa molecular weight cutoff, flash frozen, and lyophilized. Lyophilized samples were then processed for mass spectrometry analysis as described below. Four individual 100-ml culture biological replicates were prepared for each strain.

## Digestion of secretome samples

Precipitated secretomes were resuspended in 6 M urea and 2 M thiourea with 40 mM $NH_4HCO_3$ and then reduced for 1 h with 20 mM DTT. Reduced samples were then alkylated with 50 mM iodoacetamide for 1 h in the dark. The alkylation reaction was then quenched by the addition of 50 mM DTT for 15 min, and samples were digested with Lys-C (1/200 w/w) for 3 h at room temperature. Samples were diluted with 100 mM $NH_4HCO_3$ four-fold to reduce the urea/thiourea concentration below 2M, and then an overnight trypsin (1/50 w/w) digestion was performed at room temperature. Digested samples were acidified to a final concentration of 0.5% formic acid and desalted with home-made high-capacity StageTips composed of 1 mg Empore C18 material (3M) and 5 mg of OLIGO R3 reverse phase resin (Thermo Fisher Scientific) as previously described [40,41]. Columns were wet with Buffer B (0.1% formic acid, 80% acetonitrile) and conditioned with Buffer A* (0.1% TFA, 2% acetonitrile) prior to use. Acidified samples were loaded onto conditioned columns, washed with 10 bed volumes of Buffer A*, and bound peptides were eluted with Buffer B before being dried then stored at -20˚C.

## LC-MS analysis of secretome samples

Dried secretome digests were re-suspended in Buffer A* and separated using a two-column chromatography set up composed of a PepMap100 C18 20 mm x 75 μm trap and a PepMap C18 500 mm x 75 μm analytical column (Thermo Fisher Scientific). Samples were concentrated onto the trap column at 5 μl/min for 5 min with Buffer A (0.1% formic acid, 2% DMSO) and then infused into an Orbitrap Q-Exactive plus Mass Spectrometer (Thermo Fisher

Scientific) at 300 nl/min via the analytical column using a Dionex Ultimate 3000 UPLC (Thermo Fisher Scientific). 125-min analytical runs were undertaken by altering the buffer composition from 2% (0.1% formic acid, 77.9% acetonitrile, 2% DMSO) to 22% Buffer B over 95 min, then from 22% to 40% Buffer B over 10 min, and 40% to 80% Buffer B over 5 min. The composition was held at 80% Buffer B for 5 min and then dropped to 2% over 2 min before being held at 2% for another 8 min. The Q-Exactive plus Mass Spectrometer was operated in a data-dependent mode automatically switching between the acquisition of a single Orbitrap MS scan (375–1400 m/z, maximal injection time of 50 ms, an Automated Gain Control (AGC) set to a maximum of $3*10^6$ ions, and a resolution of 70k) and up to 15 Orbitrap MS/MS HCD scans of precursors (Stepped NCE of 28%, 30% and 35%, a maximal injection time of 100 ms, an AGC set to a maximum of $2*10^5$ ions, and a resolution of 17.5k).

### Proteomic analysis

Secretome samples were processed using MaxQuant (v1.6.17.0.[42]) and searched against the NCBI annotated *A. baumannii* UPAB1 proteome (NCBI Accession: PRJNA487603, 3750 proteins, downloaded 2020-3-10), a six-frame translation of the UPAB1 genome generated using the six-frame translation generator within Maxquant, and the ATCC17978 proteome (Uniprot: UP000319385, 3627 proteins, downloaded 2014-11-16) to allow the use of Uniprot annotation information associated with ATCC17978 proteins. Searches were undertaken using "Trypsin" enzyme specificity with carbamidomethylation of cysteine as a fixed modification. Oxidation of methionine and acetylation of protein N-termini were included as variable modifications, and a maximum of 2 missed cleavages were allowed. To enhance the identification of peptides between samples, the Match between Runs option was enabled with a precursor match window set to 2 min and an alignment window of 20 min with the label free quantitation (LFQ) option enabled[43]. The resulting outputs were processed within the Perseus (v1.6.0.7) analysis environment[44] to remove reverse matches and common protein contaminates prior to further analysis. For LFQ comparisons, biological replicates were grouped and data was filtered to remove any protein which was not observed in at least one group three times. Missing values were then imputed based on the observed total peptide intensities with a range of 0.3σ and a downshift of 2.5σ using Perseus. Student *t* tests were undertaken to compare the secretome between groups with the resulting data exported and visualized using ggplot2[45] within R. The resulting MS data and search results have been deposited into the PRIDE ProteomeXchange Consortium repository[46,47] and can be accessed with the identifier:PXD024736 using the Username: reviewer_pxd024736@ebi.ac.uk and Password: YS5qjHcG.

### Statistical analysis

All statistical analyses were performed using GraphPad Prism 8.0 (GraphPad Software Inc., La Jolla, CA). For all datasets, Student's unpaired *t*-tests were used.

### Supporting information

**S1 Fig. UPAB1 but not 19606 replicates in macrophages.** Representative images of infected cells at the indicated time points. Samples were stained for cell nuclei (grey), actin (magenta) and *Acinetobacter* GFP (cyan). Individual channels (greyscale) and merged images are shown. Bars: 20 μm.
(TIF)

**S2 Fig. UPAB1 replicates in multiple host cell lines.** Representative Z-projections with orthogonal views of infected J774A.1 cells (left panel) or BMDM (right panel) at 7 and 5 h p.i., respectively. Samples were stained for cell nuclei (grey), actin (magenta) and *Acinetobacter* GFP (cyan). Bars: 20 μm. White arrows indicate the vacuole.
(TIF)

**S3 Fig. UPAB1 ACVs are LC3 negative.** Representative images of J774A.1 macrophages (A) non-infected, (B) *Serratia marcescens*-infected (2 h p.i.), or (C) UPAB1-infected (1, 2.5 and 5 h p.i.). Samples were stained for cell nuclei (grey), LC3 (magenta) and bacteria expressing GFP (cyan). Individual channels (greyscale) and merged images are shown. Insets show a higher magnification of the area indicated by the white box. Bars: 20 μm.
(TIF)

**S4 Fig. UPAB1 replicates in multiple host cell lines.** Representative images of infected THP-1 (upper panels) or BMDM (lower panels) cells at 5 h p.i. Samples were stained for cell nuclei (grey), actin (magenta) and UPAB1 GFP (cyan). Individual channels (greyscale) and merged images are shown. Bars: 20 μm.
(TIF)

**S5 Fig. Recent clinical isolates replicate in J774A.1 macrophages.** Representative images of infected cells with the indicated strains. Samples were stained for cell nuclei (grey), actin (magenta) and *Acinetobacter* GFP (cyan). Individual channels (greyscale) and merged images are shown. Bars: 20 μm.
(TIF)

**S1 Table. Bacterial strains used in this study.**
(DOCX)

**S2 Table. Primers used in this study.**
(DOCX)

**S3 Table. Plasmids used in this study.**
(DOCX)

**S1 Dataset. Data used in this study.**
(XLSX)

## Acknowledgments

We thank the imaging laboratory of the Molecular Microbiology Department at Washington University in St Louis. We thank Dr. Rachel L. Kinsella for kindly providing BMDM. We thank the members of the Feldman lab for critical reading of the manuscript.

## Author Contributions

**Conceptualization:** Gabriela Sycz, Gisela Di Venanzio, Mario F. Feldman.

**Data curation:** Gabriela Sycz, Gisela Di Venanzio, Mario F. Feldman.

**Formal analysis:** Gabriela Sycz, Gisela Di Venanzio.

**Funding acquisition:** Mario F. Feldman.

**Investigation:** Gabriela Sycz, Gisela Di Venanzio, Jesus S. Distel, Mariana G. Sartorio, Nguyen-Hung Le, Nichollas E. Scott, Wandy L. Beatty.

**Methodology:** Gabriela Sycz, Gisela Di Venanzio, Jesus S. Distel, Mariana G. Sartorio, Nguyen-Hung Le, Nichollas E. Scott, Wandy L. Beatty.

**Supervision:** Gisela Di Venanzio, Mario F. Feldman.

**Validation:** Gabriela Sycz, Gisela Di Venanzio.

**Visualization:** Gabriela Sycz.

**Writing – original draft:** Gabriela Sycz, Gisela Di Venanzio, Mariana G. Sartorio, Nichollas E. Scott, Mario F. Feldman.

**Writing – review & editing:** Gisela Di Venanzio, Jesus S. Distel, Mariana G. Sartorio, Nguyen-Hung Le, Mario F. Feldman.

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
