## [Decision Letter · Decision Letter 0]

2 May 2021

Dear Dr Feldman,

Thank you very much for submitting your manuscript "Modern Acinetobacter baumannii clinical isolates replicate inside spacious vacuoles and egress from macrophages" for consideration at PLOS Pathogens. As with all papers reviewed by the journal, your manuscript was reviewed by members of the editorial board and by several independent reviewers. In light of the reviews (below this email), we would like to invite the resubmission of a significantly-revised version that takes into account the reviewers' comments.

We cannot make any decision about publication until we have seen the revised manuscript and your response to the reviewers' comments. Your revised manuscript is also likely to be sent to reviewers for further evaluation.

Sincerely,

David S. Weiss

Associate Editor

PLOS Pathogens

Denise Monack

Section Editor

PLOS Pathogens

Kasturi Haldar

Editor-in-Chief

PLOS Pathogens

orcid.org/0000-0001-5065-158X

Michael Malim

Editor-in-Chief

PLOS Pathogens

orcid.org/0000-0002-7699-2064

Reviewer's Responses to Questions

**Part I - Summary**

Reviewer #1: In this manuscript, Sycz and colleagues describe a phenomenon in which several Acinetobacter baumannii clinical isolates (such as UPAB1), unlike common laboratory stains, replicate in the vacuoles of macrophages and eventually lead to lytic cell death. The intracellular survival of UPAB1 relies on a large conjugative plasmid (pAB5) and its type I secretion system (T1SS) regulated by pAB5. The study is quite straightforward and the conclusion is justified. The importance of this study is that it urges the field to focus on the clinical stain as it clearly differs from the long and commonly used laboratory strain in the virulence and intracellular trafficking properties. The observations reported in this study will serve as a good foundation for the field to further analyze the pathogenesis mechanism of Acinetobacter baumannii, which is more relevant to understand the real-world infections caused by this important pathogen. However, the study is still of descriptive nature and lacks in-depth insights into the mechanism underlying the unique intracellular life cycle of the new Acinetobacter baumannii strain.

Reviewer #2: In this manuscript, Feldmann and colleagues demonstrate that Acinetobacter baumannii can survive intracellularly in macrophages. Previous studies with A. baumannii had largely focused on a 19606, a urinary isolate obtained in 1967. This strain does not survive in macrophages, which led to the idea that A. baumannii is an "extracellular opportunistic pathogen". The findings reported here turn this idea upside down! The manuscript is very well-written, the experiments easy to follow and the conclusions are justified. Only minor weaknesses were noted, specifically in figure presentation and lack of clarity about the prevalence of the intracellular replication phenotype amongst A. baumannii isolates.

**Part II – Major Issues: Key Experiments Required for Acceptance**

Reviewer #1: 1. I am not convinced that half of the vacuoles surrounding Acinetobacter baumannii UPAB1 have double-membrane structures, according to the right panel of Fig 1B. The lipid bilayer shown in this panel does not seem to tell the difference between the left and the right panel. More convincing data, including appropriate controls and statistics are needed to conclude this point. Also, the logic flow is a little strange. Examination of the LC3 marker on the ACVs should follow immediately after the discovery of the double-membrane structure, rather than follows the description of bacterial CFU in different macrophages. The authors should address what is the nature of the double-membrane structure as it is counterintuitively LC3-negative.

2. Which kind of lytic cell death is induced by UPAB? Markers of different cell-death types should be checked. Further, what about other clinical strains and the UPAB strain lacking pAB5 or T1SS, and do they induce similar lytic cell death? It is reasonable to speculate that the decrease in CFU of 19606 or pAB5-deficient UPAB may be due to the faster lytic death of macrophages. Therefore, more details and mechanisms should be provided.

3. Give more information about the LCP plasmid. In addition to transcriptional regulation, are there other genes involved in bacterial virulence? What are the key transcription factors? Could the macrophage replicating phenotype be reconstituted in 19606 by putting back the pAB5 plasmid?

4. A key aspect of this study is to explore the mechanism of UPAB replication in macrophages. However, data addressing this question are weak and not sufficient. More insights into the mechanism should be provided.

5. How about the life cycle of the different A. baumannii strains in epithelial cells? Is that similar to macrophages?

Reviewer #2: (No Response)

**Part III – Minor Issues: Editorial and Data Presentation Modifications**

Reviewer #1: (No Response)

Reviewer #2: (1) In Figure 2 and Figure 4, the authors show that UPAB1, 286, 398, 413 and 414 isolates are all capable of limited replication in J774A.1 macrophages. Therefore all recent clinical isolates tested here are capable of replicating inside macrophages. How representative is this phenomenon? Are there recent clinical isolates that were not able to survive and replicate for example?

(2) The confocal images do not do the findings any justice. They are too small and it is difficult to see any detail. PLOS Pathogens does not have a page limit - please make them larger. Also individual channels should be shown in greyscale, as well as an overlay of the individual channels. See a number of articles on how to present fluorescence microscopy images for color blind readers and to improve clarity.

https://journals.plos.org/plosbiology/article?id=10.1371/journal.pbio.3001161

https://www.molbiolcell.org/doi/10.1091/mbc.e11-09-0824

(3) Lines 48-51. Reference required.

(4) Line 121. Pertaining to Figure S2C. Two representative images are shown from 5 h p.i. and none from 1 h p.i. Please include an image of the early timepoint for comparison.

(5) Figure 5C. Why was LDH release not measured until 24 h p.i.? Bacterial release occurs at 5-8 h p.i., which should be coincident with LDH release. Please include LDH release data for earlier timepoints, or explain in the text why it was not done at earlier timepoints.

(6) Line 204. The Salcedo lab has recently reported similar findings in epithelial cells on BioRXiv. Please cite this article rather than "personal communication".

PLOS authors have the option to publish the peer review history of their article (what does this mean?). If published, this will include your full peer review and any attached files.

Reviewer #1: No

Reviewer #2: No
---

## [Editor Report · Decision Letter 1]

14 Jul 2021

Dear Dr Feldman,

We are pleased to inform you that your manuscript 'Modern Acinetobacter baumannii clinical isolates replicate inside spacious vacuoles and egress from macrophages' has been provisionally accepted for publication in PLOS Pathogens.

Best regards,

David S. Weiss

Associate Editor

PLOS Pathogens

Denise Monack

Section Editor

PLOS Pathogens

Kasturi Haldar

Editor-in-Chief

PLOS Pathogens

orcid.org/0000-0001-5065-158X

Michael Malim

Editor-in-Chief

PLOS Pathogens

orcid.org/0000-0002-7699-2064
---

## [Editor Report · Acceptance letter]

5 Aug 2021

Dear Dr Feldman,

We are delighted to inform you that your manuscript, "Modern Acinetobacter baumannii clinical isolates replicate inside spacious vacuoles and egress from macrophages," has been formally accepted for publication in PLOS Pathogens.

Best regards,

Kasturi Haldar

Editor-in-Chief

PLOS Pathogens

orcid.org/0000-0001-5065-158X

Michael Malim

Editor-in-Chief

PLOS Pathogens

orcid.org/0000-0002-7699-2064